# Heat shock-induced chaperoning by Hsp70 is enabled in-cell

**Drishti Guin[1], Hannah Gelman[2¤], Yuhan Wang[3], Martin Gruebele[1,2,3]***

**1** Department of Chemistry, University of Illinois Urbana-Champaign, Urbana, Illinois, United States of America, **2** Department of Physics, University of Illinois Urbana-Champaign, Urbana, Illinois, United States of America, **3** Center for Biophysics and Quantitative Biology, University of Illinois Urbana-Champaign, Urbana, Illinois, United States of America

¤ Current address: U.S. Department of Veterans Affairs, Seattle, Washington, United States of America
* mgruebel@illinois.edu

**Data Availability Statement:** All relevant data are within the manuscript and its Supporting Information files.

**Funding:** This work was supported by NSF grant MCB 1803786.

## Abstract

Recent work has shown that weak protein-protein interactions are susceptible to the cellular milieu. One case in point is the binding of heat shock proteins (Hsps) to substrate proteins in cells under stress. Upregulation of the Hsp70 chaperone machinery at elevated temperature was discovered in the 1960s, and more recent studies have shown that ATPase activity in one Hsp70 domain is essential for control of substrate binding by the other Hsp70 domain. Although there are several denaturant-based assays of Hsp70 activity, reports of ATP-dependent binding of Hsp70 to a globular protein substrate under heat shock are scarce. Here we show that binding of heat-inducible Hsp70 to phosphoglycerate kinase (PGK) is remarkably different *in vitro* compared to in-cell. We use fluorescent-labeled mHsp70 and ePGK, and begin by showing that mHsp70 passes the standard β-galactosidase assay, and that it does not self-aggregate until 50°C in presence of ATP. Yet during denaturant refolding or during *in vitro* heat shock, mHsp70 shows only ATP-independent non-specific sticking to ePGK, as evidenced by nearly identical results with an ATPase activity-deficient K71M mutant of Hsp70 as a control. Addition of Hsp40 (co-factor) or Ficoll (crowder) does not reduce non-specific sticking, but cell lysate does. Therefore, Hsp70 does not act as an ATP-dependent chaperone on its substrate PGK *in vitro*. In contrast, we observe only specific ATP-dependent binding of mHsp70 to ePGK in mammalian cells, when compared to the inactive Hsp70 K71M mutant. We hypothesize that enhanced in-cell activity is not due to an unknown co-factor, but simply to a favorable shift in binding equilibrium caused by the combination of crowding and osmolyte/macromolecular interactions present in the cell. One candidate mechanism for such a favorable shift in binding equilibrium is the proven ability of Hsp70 to bind near-native states of substrate proteins *in vitro*. We show evidence for early onset of binding in-cell. Our results suggest that Hsp70 binds PGK preemptively, prior to its full unfolding transition, thus stabilizing it against further unfolding. We propose a "preemptive holdase" mechanism for Hsp70-substrate binding. Given our result for PGK, more proteins than one might think based on *in vitro* assays may be chaperoned by Hsp70 *in vivo*. The cellular environment thus plays an important role in maintaining proper Hsp70 function.

**Competing interests:** The authors have declared that no competing interests exist.

## Introduction

The cell is a crowded environment (300–400 mg/mL of macromolecules) [1], containing many surfaces capable of transient interactions. Quinary structure (transient interactions evolved for function) and crowding (excluded volume) are features indispensable to the proper function of many cellular proteins [2,3]. On the other hand, non-specific sticking can destabilize proteins, reduce effective binding constants by competing with productive binding, or reduce the number of encounter complexes by reducing diffusion rates in the cell compared to *in vitro* [4,5].

These properties of the cytoplasm are difficult to mimic *in vitro*. The absence of an in-cell like environment is not a problem when folding of very stable proteins or very strong protein-protein interactions (nanomolar dissociation constant $K_d$) are studied in buffer, but it could have important consequences for the folding of marginally stable proteins or weak (micromolar dissociation constant $K_d$) protein-protein interactions, such as chaperoning.

Members of the 70 kDa family of heat shock proteins (Hsp70s) have long been implicated in the maintenance of in-cell protein homeostasis by binding substrates during early stages of folding or cellular stress [6–8]. Hsp70s are known to be promiscuous [9–11]. Given recent evidence that promiscuity and stress response could be tuned via quinary structure [12–14], it is possible that crowding and weak in-cell interactions could be important for proper Hsp70-substrate binding under heat shock. Hsp70 substrate affinity is foremost controlled by the binding and consecutive hydrolysis of ATP at the N-terminal nucleotide binding domain (NBD) [15]. The NBD found in the Hsp70 family shares structural similarity to actin and some sugar kinases, underscoring the importance of ATP hydrolysis machinery in the cell [16,17]. ATP hydrolysis rate is increased by substrate binding to the C-terminal substrate binding domain (SBD) followed by domain rearrangement allowing up to two orders of magnitude increase in substrate affinity [7,18].

Hsp70 has been shown to bind short peptides [19], intrinsically disordered proteins (such as RCMLA and carboxamidomethylated ribonuclease A) [6,20,21], chemically denatured proteins (such as β-galactosidase) [22], and obligate substrates (such as luciferase) [22] *in vitro*. Few examples of binding upon heat shock exist, such as NCA-SNase and staphylococcal β-lactamase exist [6,23]. These two marginally stable proteins unfold at 37˚C, and Hsp70 could only bind them in their unfolded state. Such unfolded substrates do not provide insight on how Hsp70 binding proceeds with normally folded globular proteins during heat shock.

More recent *in vitro* studies have shown binding of bacterial Hsp70 (DnaK) to full length native substrates upon force-unfolding [24]. In these studies, DnaK was not only able to stabilize unfolded proteins to prevent misfolding, but also bound near-native protein conformations to stabilize them against unfolding [25]. Significant advances have also been made in our understanding of the physical interactions between substrate and chaperone [11,26–28]. However no study has compared *in vitro* and in-cell substrate binding of a normally folded full-length protein during heat shock [6], even though heat shock is at the core of the concept of a "Heat shock protein."

To elicit a chaperoning response that mimics the cellular heat shock response, we thermally denatured PGK [29] in mammalian U-2 OS cells. Fig 1A shows a schematic of the epifluorescence microscope equipped with a 2000 nm infrared wavelength fiber laser that is used to heat the aqueous medium in and around the cell to induce a heat shock. We also showed that sticking overwhelms any productive binding of Hsp70 to its substrate phosphoglycerate kinase (PGK) *in vitro*.

Since we subject cells to sudden heat-stress in our experiments, we studied the human cytoplasmic heat-inducible Hsp70 isoform, also known as Hsp72 or HSPA1A (Table A in S1 File),

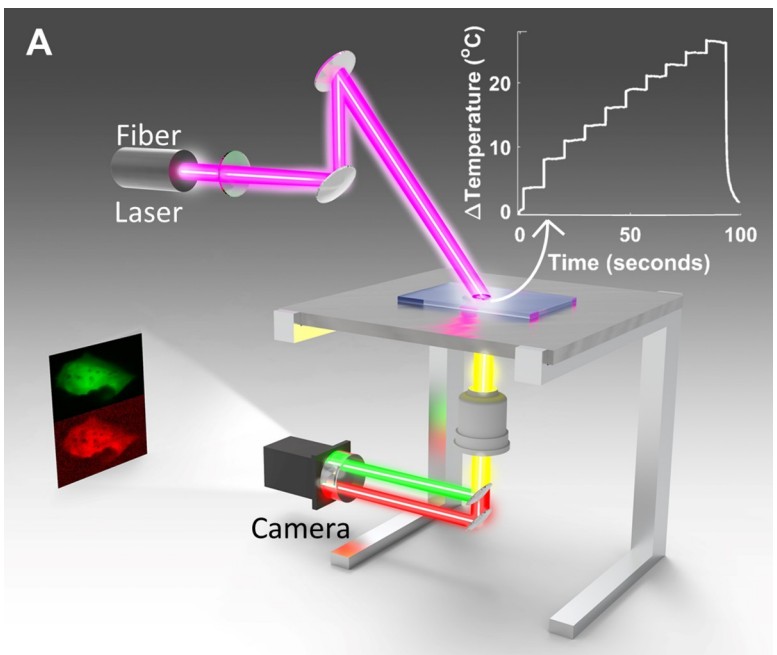

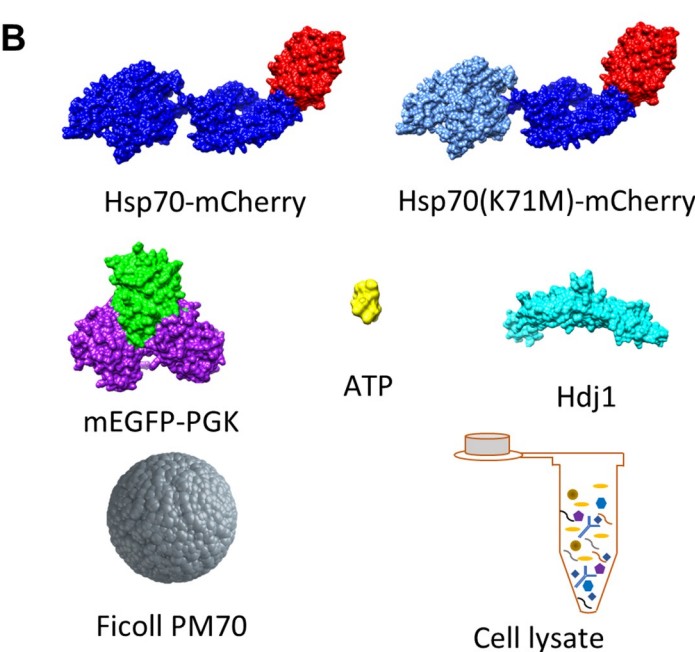

**Fig 1. Experimental design and molecular structures.** (a) Schematic of epifluorescence microscope equipped with a 2000 nm wavelength infrared fiber laser (Advalue Photonics) for fast programmed temperature ramping. (b) Structures of proteins and molecules used in experiments and Figs 4,6 and 7. Structures were accessed by either PDB IDs Hsp70 (2KHO), PGK (3PGK), Hsp40 (2QLD), mCherry (2H5Q) and mEGFP (3EVP) or PubChem ID for ATP (5957). All structures including Hsp70-mCherry were rendered using USCF Chimera 1.12.

which we refer to as Hsp70 throughout. To study binding, we engineered a FRET pair consisting of mHsp70 (Hsp70 labeled by the red acceptor mCherry at the C-terminus) and ePGK (PGK labeled by the green donor mEGFP at the N-terminus) (Fig 1B). mCherry and mEGFP

are known not to interact directly with one another below the mM level [14]. The intensity of green and red fluorescence was imaged onto a CMOS sensor as a function of temperature. Binding was detected by change in FRET efficiency between ePGK and mHsp70. Both protein concentrations were in the few micromolar range in-cell, comparable to physiological concentrations of these proteins in mammalian cells.

This paper shows in five steps that the cellular environment is needed for an effective chaperoning response. 1) We first show that fluorescently labeling PGK and Hsp70 does not significantly disrupt their stability and function, 2) we show that denaturant-unfolded PGK refolding by Hsp70 *in vitro* is no different than for an ATPase activity-deficient mutant, 3) we show that Hsp70 upon heat shock *in vitro* does not bind to PGK any differently than the same ATPase activity-deficient mutant, and that crowding or co-factors alone do not rescue this deficiency, 4) we show that an ATPase-dependent heat shock response does occur in mammalian cells, and 5) we show that the in-cell response has an onset even below the melting temperature of PGK.

Thus Hsp70 acts as an ATP-specific chaperone for PGK in-cell, but not under denaturant or heat-shock conditions *in vitro*. There is a precedent for such "pre-emptive holdase" activity for the Hsp70 homolog DnaK, where the chaperone binds to full-length native proteins in force-unfolding experiments [24]. Our experiments establish such a mechanism for Hsp70 and its substrate PGK in cells.

## Materials and methods

### Protein expression and purification

Protein expression was carried out according to previously established protocol [30]. Fusion protein sequences were cloned into pDream 2.1/MCS vector by Genscript Corp. and used for dual expression in E. coli and mammalian cells unless stated otherwise. Sequences of relevant Hsp70 (HSPA1A), Hsp40 (Hdj1) and PGK mutants are listed in the Table A in S1 File. Hsp70 was cloned with a C-terminal mCherry (mHsp70) and mEGFP (mEGFP-Hsp70) fusion protein with a 6×His-tag for purification. Hsp40 sequence with 6×His tag was cloned into pET15b for expression in *E. coli*. All PGK sequences were also cloned into pDream as N-terminal mEGFP fusion proteins with 6xHis-tags for purification purposes (called 'ePGK' here). PGK mutants (ePGK0-ePGK3) with varying stabilities were designed in-house using PCR site-directed mutagenesis. The goal of the fluorescent labeling was not to obtain a maximum FRET signal when Hsp70 and PGK interact, but to obtain enough signal while not disrupting the function of the proteins.

For protein expression, the plasmid of interest was chemically transformed into BL21-CodonPlus(DE3)-RIPL cells (Agilent) according to manufacturer protocol. Transformed cells were grown in 1L Lennox LB broth (Fisher Biosciences) at 37˚C for 4 hours. The cells were then induced with 1 mM isopropyl β-d-1-thiogalactopyranoside (IPTG, Inalco) and allowed to express protein for 12–16 h at 18˚C. The cells were pelleted by centrifugation (Beckman Coulter Avanti J-E, 5000 rpm, 25 minutes, 10˚C) and resuspended in lysis buffer (500 mM NaCl, 50 mM Na2PO4, 20 mM imidazole, pH 7.4). The cell suspension was then supplemented with protease inhibitors (1 mM PMSF, Sigma) and 20 μL DNaseI (New England Biolabs)/10 mL suspension. Cells were then broken by sonication (Qsonica, 70% intensity, 6 s pulse/min, 6 min total process time). The cell lysate was re-centrifuged (10000 rpm, 20 min, 10˚C) to remove debris and filtered once through a 0.45 μm filter (Millipore) and twice through 0.22 μm filter (Millipore).

Each protein was isolated by affinity chromatography. Proteins were purified on a 5-mL HisTrap HP column by FPLC (GE Healthcare Life Sciences AKTA) and eluted with 500 mM

NaCl, 50 mM $Na_2HPO_4$ and 500 mM imidazole; pH 7.4. Protein purity was assessed using SDS-PAGE gel electrophoresis, mass spectroscopy and protein was dialyzed into appropriate storage buffers: 1) PGK storage buffer: 20 mM phosphate, pH 7 and 2) Hsp70/Hsp40 storage buffer: 100 mM Tris-HCl, 0.1 mM EDTA, 100 mM NaCl, pH 6.9 with KOH. For long term storage, glycerol was added to a final concentration of 30% volume/volume, flash frozen in liquid $N_2$ and stored at -70˚C.

### Stopped flow PGK refolding assay

To monitor PGK unfolding and refolding, a FRET-labeled PGK1 (FRET-PGK1) was designed with an N-terminal mEGFP and a C-terminal mCherry label and cloned into a pDream vector by Genscript [31]. Stopped flow measurements were conducted with an Olis RSM 1000 instrument. FRET-PGK1 (efPGK1) was excited at 475 nm and emission spectra were collected between 493 to 720 nm at 10 scans/sec scan speed. Stopped flow unfolding of efPGK1 was measured by rapid mixing of 42 μM efPGK1 and 0.6 M GuHCl in a 1:5 ratio. The final concentration after mixing for unfolding experiments was 7 μM efPGK1 and 0.5 M GuHCl. Stopped flow refolding of efPGK1 was conducted by rapidly diluting 42 μM efPGK1 denatured in 0.5 M GuHCl in buffer with or without Hsp70/Hsp40 in a 1:5 ratio. The final concentration after mixing for refolding experiments was 7 μM efPGK1 and 0.083 M GuHCl. Chaperone free refolding buffer contained 2 mM ATP and 1 mM DTT in K1 buffer (25 mM HEPES, 50 mM KCl and 10 mM $MgCl_2$; pH 7.6 with KOH). Chaperone assisted refolding was conducted in K1 buffer with 2 mM ATP, 1 mM DTT, 100 μM Hsp70/Hsp70K71M and 22.5 μM Hsp40.

### U-2 OS cytoplasmic lysate preparation

Cells were grown to 70% confluence in T-75 culture flasks (Thermo Fisher Scientific). Prior to lysis, growth medium was removed, and cell monolayer was washed with 10 mL PBS (Corning). Cells were trypsinized (Corning) with 1 mL trypsin/flask and collected in 15 mL conical tubes. The cells were then pelleted by centrifuging for 5 minutes at 300×g (NuAire Awel C48). The supernatant was aspirated, and the cells were washed with 10 mL ice-cold PBS. The pellet was spun down again for 5 minutes at 300×g after which the PBS wash was aspirated. A 1 mL aliquot of lysis buffer was prepared by mixing cold Pierce lysis buffer (Thermo Fisher Scientific, 87787) was supplemented with 10 μL 100× protease inhibitor cocktail (Thermo Fisher Scientific, 78442). The pellet was resuspended in 200 μL ice-cold lysis buffer and incubated on ice for 30 minutes. The lysed cells were centrifuged again for 5 minutes at 13000 g on a tabletop centrifuge (Eppendorf 5415D) and the supernatant was collected. The protein concentration of the lysate was estimated by a BCA assay (Thermo Fisher Scientific, 23225). Aliquots of cell lysate were stored at -20˚C.

### *In vitro* fluorimeter melts

Prior to all measurements, glycerol was removed from the frozen stocks by spin filtration buffer exchange. Tryptophan fluorescence measurements and *in vitro* FRET binding experiments were conducted on an FP8300 spectrofluorometer equipped with Peltier temperature control (JASCO). Tryptophan was excited at 295 nm, and emission spectra were collected from 290 to 450 nm. Generally, our samples with ATP are excited at 295 nm to avoid the intense ATP absorbance at 280 nm (peak excitation wavelength for tryptophan). Emission is collected from 290 nm which overlaps with the excitation. Generally, this is done so we can monitor excitation peak for signs of aggregation. Formation of aggregates generally leads to scatter and a drastic decrease in excitation signal. If this occurs the experiments are repeated. Samples were measured in 300 μL cuvettes at 5 μM concentrations, unless otherwise noted.

For all FRET measurements mEGFP was excited at 485 nm and emission spectra were collected from 480 to 700 nm in 300 μL cuvettes. Unfolding and aggregation of Hsp70 was monitored by the change in FRET efficiency *vs*. temperature by melting an equimolar mixture of mHsp70 (1 μM) and mEGFP-Hsp70 (1 μM) with and without 2 mM ATP and 10 mM DTT in K1 buffer.

For *in vitro* FRET binding experiments ePGKn (n = 0–3) was heated with mHsp70 with 10 mM DTT and 2 mM ATP in K1 buffer. For measurements with 1:1 mHsp70:ePGK, 1 μM ePGK was mixed with 1 μM mHsp70 and for 5:1 mHsp70:ePGK, 5 μM mHsp70 was mixed with 1 μM ePGK. ATP was maintained in all *in vitro* experiments by addition of an ATP regenerating mix. The mix consists of 50 mM creatine phosphate, 0.2 mg/mL of creatine phosphokinase in appropriate buffer. Regeneration of ATP has been shown to increase refolding of denatured luciferase by Hsp70 [32]. For crowding Ficoll70 (Sigma) was added to a final concentration of 300 mg/mL along with 1 μM Hsp40. For measurements in cell lysate and Ficoll70, 0.6 mg/mL cell lysate was added in addition to 300 mg/mL Ficoll70 and 1 μM Hsp40. Temperature melts with Ficoll showed signs of aggregation at higher temperatures.

## β-galactosidase Hsp70 refolding assay

The refolding assay was performed according to a previously described protocol [22]. Briefly 1 μM β-galactosidase was denatured by 10-fold dilution into denaturation buffer (K1 buffer supplemented with 5 mM BME) with 3 M GuHCl. For a folded control measurement β-galactosidase was diluted into denaturation buffer without GuHCl. Both denatured and folded β-galactosidase were then incubated at 30°C for 30 minutes. Refolding was then initiated by 125-fold dilution of denatured β-galactosidase into refolding buffer (K1 supplemented with 10 mM DTT and 2 mM ATP) supplemented with BSA (3.2 μM) or chaperone (1.6 μM Hsp70 or 3.2 μM Hsp40) at 4°C. For a folded control, folded β-galactosidase was also diluted 125-fold into refolding buffer with 3.2 μM BSA. At each time-point 10 μL of the appropriate mixture was removed and added to 10 μL 0.8 mg/mL ONPG substrate. The ONPG was then incubated at 37°C for 15 minutes. The chromogenic reaction was arrested by adding 80 μL 0.5 M sodium carbonate. The extent of chromogenic reaction was estimated by measuring absorbance at 412 nm in a UV-Vis spectrophotometer (Shimadzu UV-1800) in a quartz cuvette. Refolding activity of Hsp70 was estimated by calculating the percentage recovery of GuHCl denatured β-galactosidase with respect to folded control.

## FRET binding experiments in cells

U-2 OS cells (ATCC) were grown in DMEM (Corning) supplemented with 10% FBS, 5% penicillin streptomycin (Fisher) and 5% sodium pyruvate (Fisher). At ~70% confluence cells were transferred on to pre-cleaned glass coverslips (18 mm × 18 mm, #1.5) in a 35 mm falcon dish (MatTek Corp.) by trypsinization. They were then co-transfected with the appropriate plasmid (s) with Lipofectamine 2000 (Fisher) (5 μL Lipofectamine for every 2 μg plasmid) in DMEM without penicillin. 4 hours after transfection cells were washed with PBS and allowed to grow for 18–26 hours in DMEM with penicillin. For co-transfection plasmid concentrations were optimized for 1:1 expression of both proteins. For mHsp70 and ePGK roughly 2 μg mHsp70 and 1 μg ePGK was used for co-transfecting 3 35 mm falcon dishes. For mCherry/ePGK and mHsp70/mEGFP controls roughly 2 μg mHsp70/ePGK and 1 μg mCherry/mEGFP was used for co-transfecting 3 35 mm falcon dishes.

18 to 26 hours after transfection coverslips were washed with PBS (Corning) and adhered to a slide using 120 μm thick spacers (Grace Bio-Labs) for imaging. Cells were imaged in Opti-MEM (Fisher) supplemented with 15% FBS. mEGFP in the cells was excited by a white LED

by passing the light through a Chroma ET470/40x bandpass filter and mCherry was excited through a ET580/25x bandpass filter [30]. Both mEGFP and mCherry emission was monitored on a Lt225 camera equipped with a CMOS sensor (Lumenera).

Prior to the experiment, each fluorescent protein was individually excited to obtain intensities at room temperature. Hsp70 and PGK were assumed to be in the unbound state at room temperature. Initial concentrations of the two proteins was calculated from their individual fluorescence intensities using a predetermined *in vitro* calibration. Only cells that expressed roughly equimolar quantities of mEGFP and mCherry labeled proteins were chosen for experiments. Cells were heated using fast temperature jumps between 19–46°C and imaged at 60 fps using LabVIEW (National Instruments) [30].

## Data analysis

All data was analyzed using MATLAB (MathWorks). *In vitro* tryptophan fluorescence measurements were analyzed by monitoring wavelength peak shift. CD spectra were analyzed by calculating the change in the CD mean residue ellipticity (MRE) at 222 nm [33].

Both *in vitro* and in-cell FRET binding measurements were monitored using FRET efficiency ($E_{FRET}$). The $E_{FRET}$ was calculated according to:

$$E_{FRET} = \frac{mCherry\ (acceptor)\ intensity}{mEGFP\ (donor)\ intensity + mCherry\ intensity} \quad [1]$$

*In vitro* donor and acceptor intensity were calculated by measuring area under the curve between 495–585 and 585–700 nm, respectively. In-cell binding was detected via FRET from mEGFP to mCherry and quantified as described previously in [31]. FRET efficiency was estimated at fourteen equally spaced temperatures from 20–46°C by interpolation and normalized to the first temperature point. The traces were then corrected for bleaching by fitting a line through the first three data points and subtracting the fitted slope. The corrected traces were then averaged.

Melting temperature ($T_m$) or binding temperature ($T_0$) were calculated using a two-state sigmoidal fit to the experimental data. If binding is driven by unfolding, sigmoidal curves are also appropriate for the bimolecular process. Native and denatured state baselines were assumed to be linear as a function of temperature, so that the signal $S$ can be estimated by:

$$S_i = m_i(T - T_X) + b_i, \quad [2]$$

where $i$ is either the native or substrate-chaperone unbound state (N) or the denatured or substrate-chaperone aggregated state (D), $m$ is the slope and $b$ is the intercept. $T_X$ is the corresponding midpoint of the curve ($T_m$ or $T_0$) with the temperature $T$. The total signal $S(T)$ can then be estimated as a sigmoidal function with respect to temperature $T$:

$$S(T) = S_N(T)f_N(T) + S_D(T)f_D(T), \quad [3]$$

where, $f_N$ and $f_D$ are the populations of the N and D states, respectively.

$$K_{eq} = e^{-\frac{\Delta G_{N \to D}}{RT}},$$
$$\Delta G_{N \to D} \approx \delta g_1(T - T_X), \quad [4]$$
$$f_N = \frac{K_{eq}}{1 + K_{eq}},\ \text{and}\ f_D = \frac{1}{1 + K_{eq}}$$

Fits to Eq 2 yield the melting temperature ($T_m$) or binding temperature ($T_0$). For a three-

state sigmoidal fit the total signal $S(T)$ can then be estimated with respect to temperature $T$ as:

$$S(T) = S_U(T)f_U(T) + S_B(T)f_B(T) + S_A(T)f_A(T),$$ [5]

where $U$ is the substrate-chaperone unbound state, $B$ is the substrate-chaperone bound state and $A$ is the substrate-chaperone aggregated state. PGK self-aggregates on unfolding (Supplementary results and Fig A in S1 File). However, Eq 5 is fit to the FRET between Hsp70 and PGK. The three states which would contribute to this FRET would be the unbound low-fret state, bound mid-fret state and aggregated high-fret state. Even though some PGK is likely self-aggregated at high temperatures after unfolding this is FRET invisible and hence has not been accounted for in Eq 5. $S_U$, $S_B$ and $S_A$ can be modeled by Eq 2. $f_U$, $f_B$, and $f_A$ are the populations of the unbound, bound and aggregated states, respectively.

$$K_{eq1} = e^{-\frac{\Delta G_{U \to B}}{RT}},$$

$$K_{eq2} = e^{-\frac{\Delta G_{B \to A}}{RT}},$$ [6]

$$\Delta G_{U \to B} \approx \delta g_1(T - T_{01}), \text{ and } \Delta G_{B \to A} \approx \delta g_1'(T - T_{02}),$$

where $T_{01}$ is the binding temperature midpoint for substrate and chaperone and $T_{02}$ is the aggregation temperature midpoint where both chaperone and substrate are unfolded above 50°C and aggregate.

$$f_U = \frac{K_{eq1}}{1 + K_{eq1} + K_{eq1}K_{eq2}},$$

$$f_B = \frac{K_{eq1}K_{eq2}}{1 + K_{eq1} + K_{eq1}K_{eq2}},$$ [7]

$$f_A = \frac{1}{1 + K_{eq1} + K_{eq1}K_{eq2}}$$

For stopped flow experiments, FRET efficiency was calculated the same as shown in Eq 1. Donor and acceptor intensity were calculated by measuring area under the curve between 535–585 nm and 585–635 nm, respectively. The unfolding and refolding curves were fit to Eqs 8 and 9, respectively as shown below.

$$ae^{-k_1 t} + be^{-k_2 t}$$ [8]

$$\frac{ae^{-kt}}{2} - \frac{a}{2}$$ [9]

While unfolding was fit to a double exponential Eq 8 which accounts for a fast unfolding phase and a slow unfolding phase, the refolding was just fit to a single exponential Eq 9.

## Results and discussion

### mCherry labeling only slightly reduces Hsp70 activity and stability *in vitro*

Although C-terminal fluorescent protein fusions of Hsp70 have been shown to be functional previously [34], fluorescent protein tags such as mCherry could perturb stability and function of the tagged Hsp70 [35]. To confirm that labeled mHsp70 retains its refolding activity, we conducted standard refolding assays with β-galactosidase [22] (Fig 2A). Wild-type Hsp70 (wt-

Hsp70) with co-chaperone Hsp40 recovered about 55% of unfolded β-galactosidase, while mHsp70 with Hsp40 recovered about 35%.

Hsp70 function is modulated by ATPase activity and mutants lacking ATPase activity should show diminished substrate recovery. A lysine at position 71 (K71) in the nucleotide binding domain is essential for ATP hydrolysis and any mutations at that position abrogate ATPase activity [36]. The ATPase activity for mHsp70K71M was tested, and the mutant showed no ATPase activity compared to mHsp70 (Supplementary Results and Fig B in S1 File). As expected, mHsp70K71M also showed much lower (<15%) β-galactosidase recovery, whether Hsp40 was present or not. mCherry, Hsp40 and the model crowder protein BSA showed no significant chaperoning activity on their own.

Surprisingly, mHsp70 without the co-chaperone Hsp40 recovered about ~28% of unfolded β-galactosidase, refolding almost twice as much β-galactosidase as wt-Hsp70 by itself (~15% recovery). We speculate that mCherry linked to Hsp70 helps with substrate recruitment due to crowding or hydrophobic surface patches on the mCherry surface. In effect, mCherry acts as a low-performance co-chaperone substitute.

To confirm that the stability of the labeled construct is the same as wt-Hsp70, thermal stability of mHsp70 was characterized in a temperature-controlled fluorimeter and compared to published results. The thermal stability of Hsp70 has been previously characterized *in vitro* where Hsp70 unfolded with a melting temperature ($T_m$) of 42°C without ATP and 60°C with $Mg^{2+}$/ATP [37]. This unfolding both with and without ATP was accompanied by a 25% structure loss (by circular dichroism (CD)). Using FPLC gel-filtration the authors also showed that this unfolding transition led to self-aggregation and formation of oligomeric species.

We monitored Hsp70 unfolding and aggregation by tracking FRET efficiency ($E_{FRET}$) change of an equimolar mixture of donor-Hsp70 (mEGFP-Hsp70) and acceptor-Hsp70 (mHsp70) *in vitro*. Both mCherry and mEGFP are stable in the temperature range tested and do not unfold until ≥70°C (Fig C in S1 File). Thermal melts of mHsp70 and mEGFP-Hsp70 showed the expected increase in $E_{FRET}$ upon unfolding due to self-aggregation. The $E_{FRET}$ curves fit to a $T_m$ of 45°C without ATP, and 55°C with ATP (Fig 2B) similar to previously published results. mHsp70 unfolding was also characterized by CD and tryptophan fluorescence, and showed similar unfolding behavior with the different spectroscopic probes (Table B in S1 File). These changes in FRET at high temperature are not to be confused with productive binding of Hsp70 to substrates.

## Four GFP-labeled PGK mutants 'ePGK' have successively lower melting points

*E. coli* PGK is a known substrate of the *E. coli* Hsp70 (DnaK) based on cell-extract results [29]. We used yeast PGK for our experiments. Yeast PGK and human PGK share 65% sequence identity and 76% sequence similarity (Emboss Needle alignment tool). We designed and studied four different mutants of PGK labeled with mEGFP (Table C in S1 File) with successively lower thermal denaturation midpoints $T_m$ to map out the relationship between substrate stability and chaperone binding. The thermal denaturation of these 'ePGK' mutants was probed by tryptophan fluorescence peak shift or CD mean residue ellipticity (Fig D in S1 File). Mutant ePGK0 has a $T_m$ (48.4°C) well above the viable temperature range for mammalian cells. It serves as a control where little binding is expected below 48°C. ePGK1 ($T_m$ = 45.8°C), ePGK2 ($T_m$ = 43.3°C) and ePGK3 ($T_m$ = 40.5°C), although based on very different mutations (see SI), all have a $T_m$ that lies in the range where unfolding is accessible in the cell. All mutants have $T_m$s above physiological temperature for the U-2 OS human cell line.

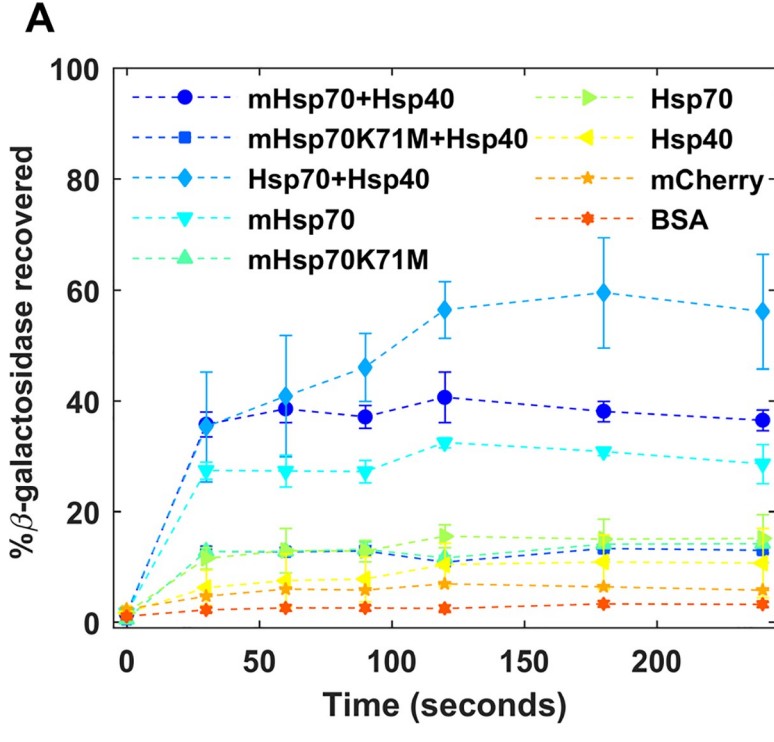

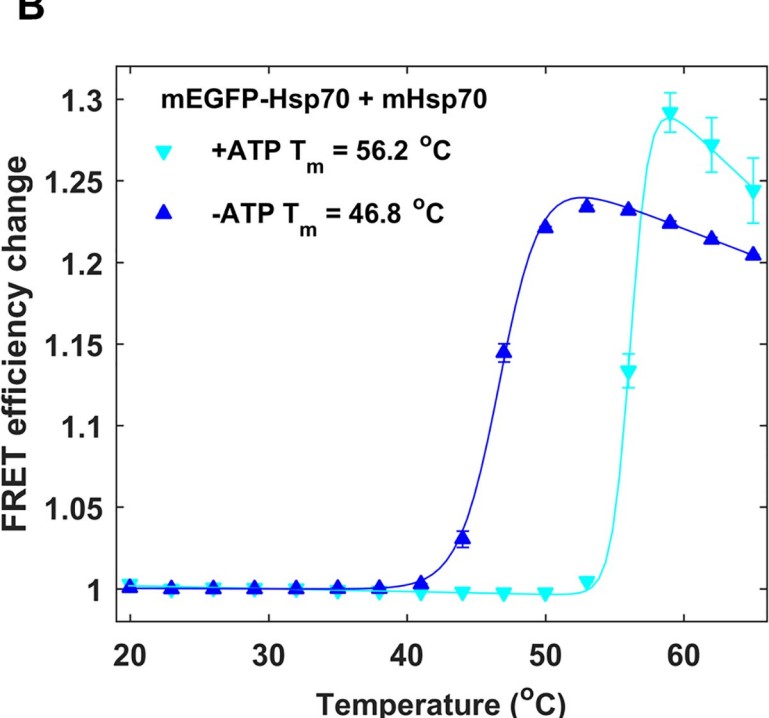

**Fig 2. *In vitro* characterization of mHsp70.** Filled markers and dashed lines show experimental data and solid lines are fits to experimental data. Error bars are standard error from mean for two measurement repeats. (a) β-galactosidase assay to assess refolding activity of mHsp70 compared to wt-Hsp70 and mHsp70K71M. (b) Unfolding of mHsp70 probed by FRET as a function of temperature with and without ATP. FRET was monitored between mEGFP-Hsp70 and mHsp70. Errors in $T_m$ reflect the standard deviation of $T_m$ values obtained from individually fitting two measurements.

## mHsp70 is not an ATP-dependent chaperone for PGK *in vitro* upon refolding from denaturant

To see whether ePGK is an ATP-specific *in vitro* substrate of human Hsp70, we performed stopped-flow PGK refolding assays. PGK1 was labeled with mEGFP at the N-terminus and mCherry at the C-terminus [38] such that refolding can be monitored by the change in FRET efficiency between mEGFP and mCherry as the two termini move closer together. The refolding rate of initially GuHCl-denatured FRET-PGK1 in buffer and buffer supplemented with Hsp70/Hsp40 was compared. Optimal GuHCl concentrations for refolding experiments were determined by performing isothermal titrations of FRET-PGK1 with GuHCl, yielding a midpoint concentration ($C_m$) of 0.31 M (Fig E in S1 File) in agreement with previously reported values [39].

Unfolding of efPGK1 in 0.5 M GuHCl showed a fast phase with $\tau_1 \approx 5.1 \pm 0.2$ secs and a slow phase with $\tau_2 \approx 80.7 \pm 1.5$ secs (Fig 3). Addition of both Hsp70 and Hsp40 improved folding efficiency of efPGK1 from 47% without Hsp70 to 57% with Hsp70. However, replacement of Hsp70 by the ATP-ase activity-deficient mutant Hsp70K71M yielded the same improvement. Even though these stopped flow experiments show that Hsp70 improves efPGK1 refolding, this activity is ATP-independent. Thus, while PGK is a substrate of Hsp70, its interaction *in vitro* when refolding from denaturant is non-specific.

## mHsp70 is not an ATP-dependent heat shock chaperone for PGK *in vitro*

To study chaperoning of ePGK0-3 by mHsp70 under heat shock *in vitro*, thermal scans were FRET-detected in buffer K1 with excess ATP (see Methods). 5:1, 1:1 and 1:5 concentration ratios of substrate and chaperone were used. In order to differentiate between melting and association, we refer to the protein thermal denaturation midpoint as $T_m$, and the association midpoint as $T_0$. Fig 4 shows the interaction of 5× excess mHsp70 (5 μM) with ePGK0-3 (1 μM). In ATP buffer, $E_{FRET}$ *vs.* temperature shows a cooperative transition with an onset of 49˚C and a temperature midpoint ($T_0$) of 55˚C using a two-state sigmoidal fit (see Methods). This $T_0$ is similar to the $T_m$ of mHsp70 with ATP in Fig 2B, and high temperature association is thus aggregation of unfolding ePGK0 with unfolded mHsp70, and not productive chaperone-substrate binding.

A transition with a lower onset at 35˚C was clearly observed in Fig 4 for ePGK3 (35˚C). Control experiments with mHsp70+mEGFP as well as mCherry+ePGK0-3 showed no such increase in FRET efficiency $E_{FRET}$ (Fig F in S1 File). Thus, the association is not of mEGFP with mHsp70 or mCherry with PGK. The $E_{FRET}$ curve for ePGK3 and mHsp70 could be fit using a three-state model with two $T_0$s ($T_{01}$ and $T_{02}$). $T_{01} = 40$˚C agrees with the $T_m$ for ePGK3. This initial transition is due to association of mHsp70 and ePGK3 as it unfolds. $T_{02}$ fit to 54˚C and corresponds to the aggregation of unfolded mHsp70 and ePGK3 as discussed above for ePGK0. The binding curves for ePGK1 and ePGK2 could not be fitted reliably by a three-state model but also showed some early onset.

We also tested binding at other mHsp70:ePGK ratios, most importantly for comparison with our in-cell mHsp70:PGK ratio of $\approx 1{:}1$. At 1:1 or 1:5 mHsp70:ePGKs (Fig G in S1 File), no binding is detectable at all *in vitro*.

We hypothesized that the earlier onset of Hsp70-PGK3 interaction at the 5:1 ratio might signal productive binding, which requires ATP-dependence. Such substrate binding would be abolished for the ATPase-deficient mutant mHsp70K71M. However, 5:1 mHsp70K71M:ePGK3 binding yielded curves like those observed with mHsp70 (Fig 5A). We conclude that our hypothesis of productive *in vitro* binding is falsified and that mHsp70 does not act as an

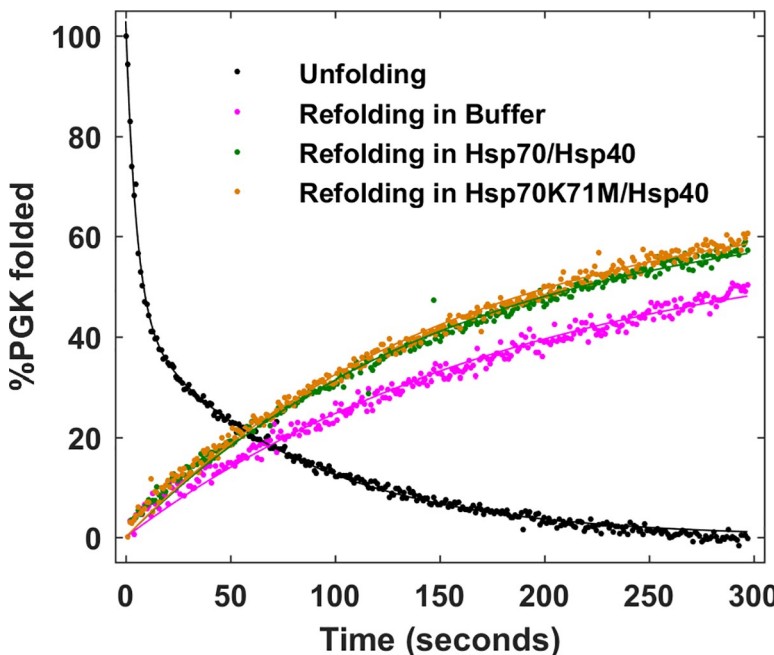

**Fig 3. *In vitro* FRET-PGK1 refolding assay with Hsp70.** Markers show experimental data and solid lines show fits to experimental data. Data displayed was binned at 1 second time intervals.

ATP-dependent heat shock chaperone for ePGK3. As the substrate protein unfolds, it simply sticks *in vitro* to the Hsp70 chaperone in an ATPase activity-independent manner.

Any productive binding of mHsp70 to ePGK3 should show an $E_{FRET}$ change above the background (~5%) observed for ATP-independent interaction. We next checked whether addition of 10 μM co-chaperone Hsp40 would increase $E_{FRET}$ above the background. The addition of Hsp40 did not significantly increase mHsp70-ePGK3 binding, and mHsp70K71M again showed a similar curve with Hsp40. We tested various concentrations of co-chaperone Hsp40 to 5:1 mHsp70:ePGK3, but no significant increase over the background could be observed at any of the concentrations tested (Fig H in S1 File). Thus, the co-chaperone Hsp40 *in vitro* does not rescue ATP-independent sticking of PGK and Hsp70 to become ATP-controlled productive binding.

Since the cellular environment is extremely crowded, we also conducted experiments *in vitro* with an artificial crowding agent, 300 mg/mL Ficoll70, at an excluded volume similar to the cytoplasm. We again tested 1:1 (Fig 6A), 1:5 (Fig I in S1 File) and 5:1 mHsp70:ePGKs (Fig I in S1 File). Ficoll70 resulted in two well-resolved cooperative transitions for ePGK2 and 3. We discuss 1:1 binding in more detail here because this condition is the closest to our in-cell experiments. The binding curves for ePGK2 and 3 were fit using a three-state model and both yielded a $T_{02}$ of ~54°C (aggregation of unfolded mHsp70 and unfolded ePGK). $T_{01}$ fit to 43.2°C for ePGK2 and 40.0°C for ePGK3, both in good agreement with the $T_m$ for ePGK2 and ePGK3. Therefore, this early transition is due to the association of mHsp70 to ePGK as it unfolds. ATPase-deficient mHsp70K71M:ePGK controls in Ficoll70 (Fig 5C) again showed the same change in $E_{FRET}$, with similar $T_{01}$. Thus, binding in a crowded *in vitro* environment is still dominated by ATP-independent sticking of unfolded substrate to the chaperone.

To mimic the cell more closely, we also performed experiments in 0.6 mg/mL cell lysate plus buffer containing 300 mg/ml of the crowding agent Ficoll70 (Fig 6B) with 1:1 mHsp70:ePGK. This significantly reduced sticking of ePGK3 compared to just Ficoll70, from ~10% to

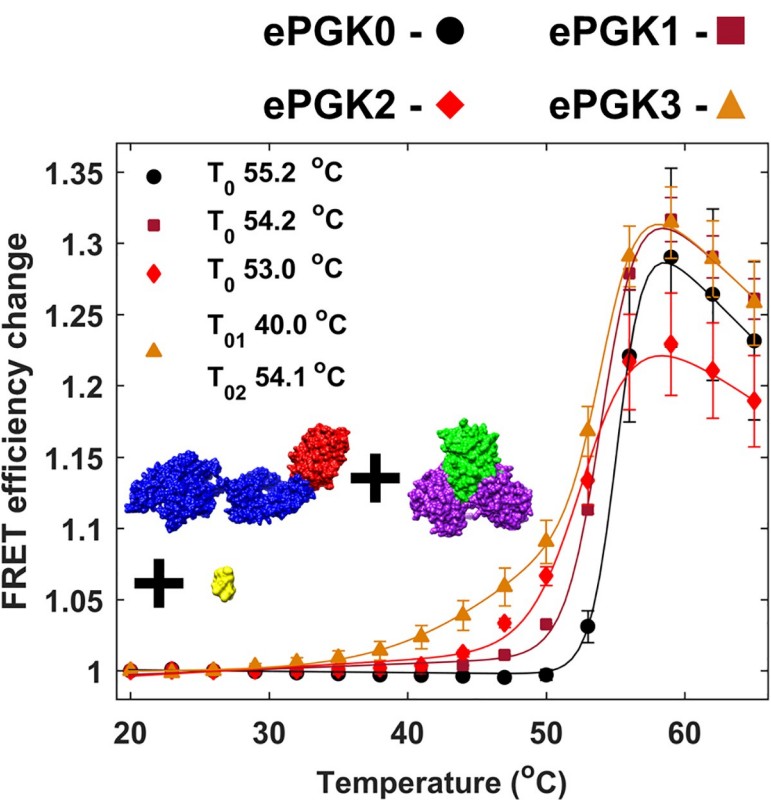

**Fig 4. *In vitro* binding of 5:1 mHsp70 and ePGKs with ATP.** Colors are ePGK0 (black), ePGK1 (dark-red), ePGK2 (red) and ePGK3 (orange). Filled markers show experimental data and solid lines are fits to experimental data. Error bars are standard error from mean for two measurement repeats. Errors in $T_m$ reflect the standard deviation of $T_m$ values obtained from individually fitting three measurements.

~5%. Sticking was abolished completely for ePGK2. We hypothesize that the combination of crowding and weak interactions of lysate osmolytes and macromolecules with Hsp70 and

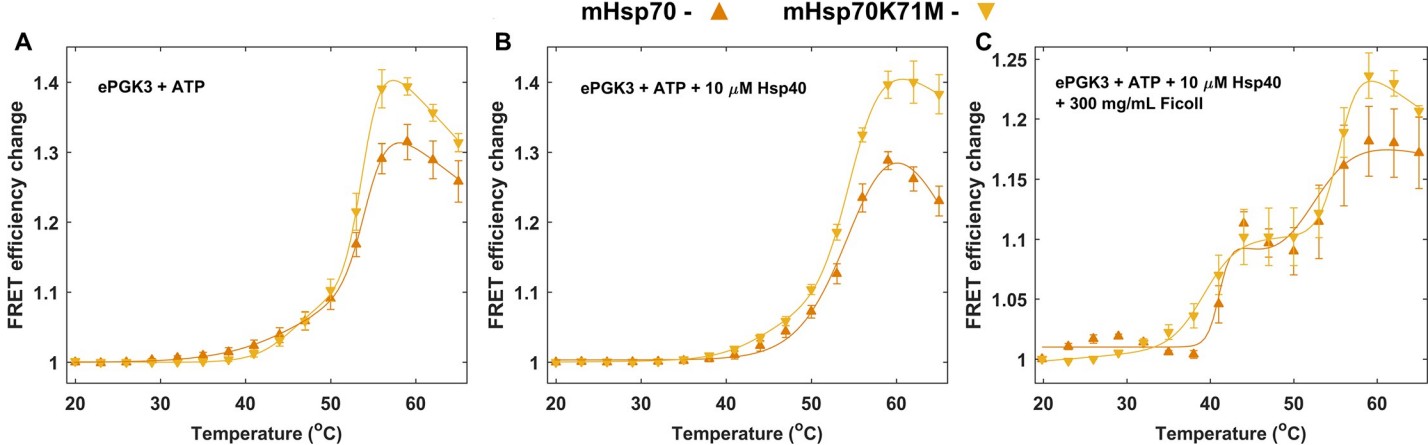

**Fig 5. *In vitro* binding of 5:1 mHsp70 and ePGK3** with (A) ATP, (B) ATP and 10 μM Hsp40 and (C) ATP, 10 μM Hsp40 and 300 mg/mL Ficoll70. Filled markers show experimental data and solid lines are fits to experimental data. Error bars are standard error from mean for two measurement repeats. Both mHsp70 and mHsp70K71M show similar binding curves.

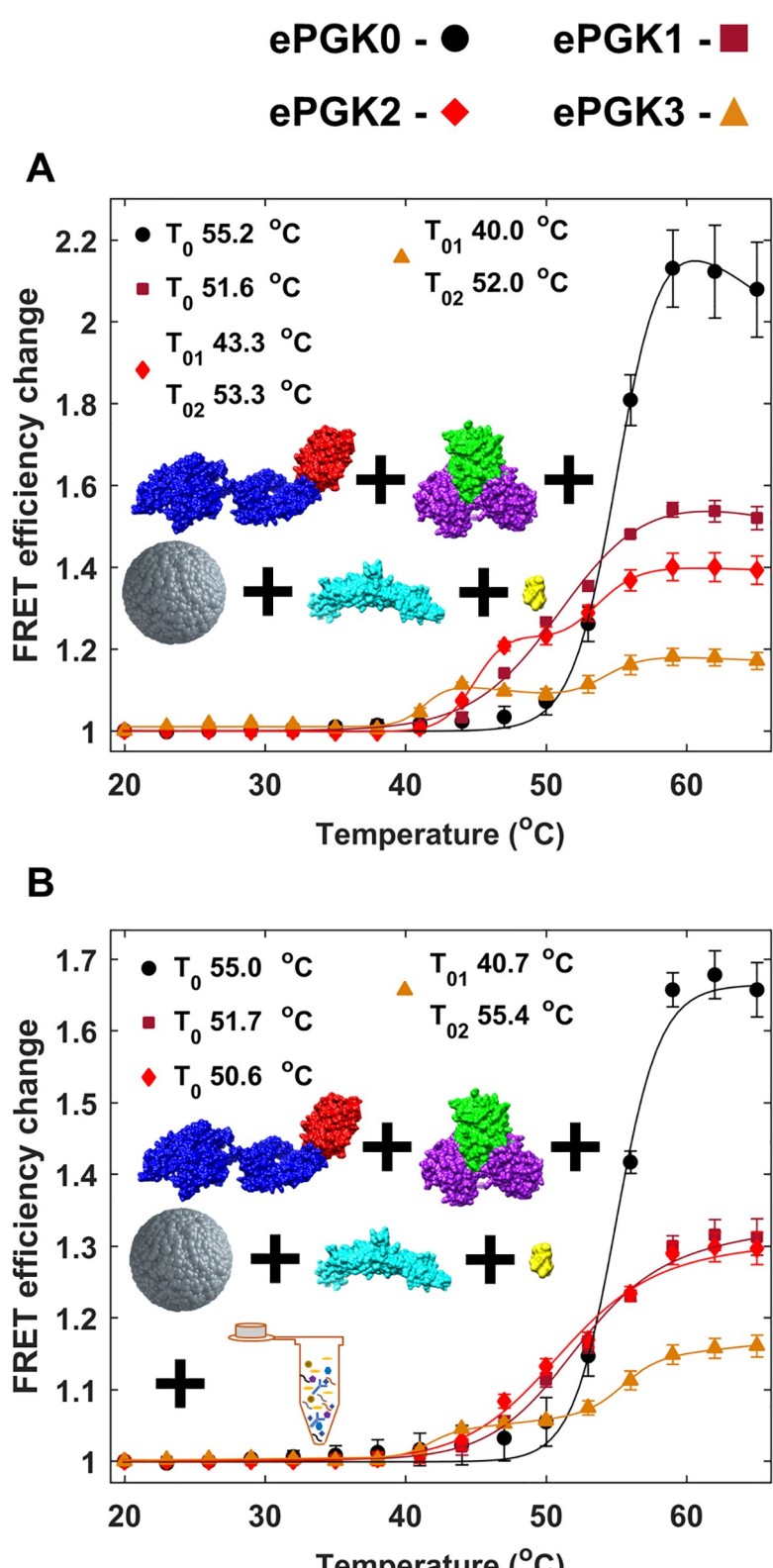

**Fig 6.** *In vitro* **binding of 1:1 mHsp70 and ePGKs with and without cell lysate.** Filled markers show experimental data and solid lines are fits to experimental data. Error bars are standard error from mean for two measurement repeats. Fit errors reflect the 1 standard deviation precision of the fit. Binding of ePGK0 (black), ePGK1 (dark red), ePGK2 (red) and ePGK3 (orange) in buffer with (A) Ficoll70, ATP and Hsp40 with mHsp70 without cell lysate and (B) Ficoll70, cell lysate, ATP and Hsp40 with mHsp70.

PGK reduces sticking and could promote proper heat shock function of Hsp70 in cells. All binding $T_0$s are reported in Table D in S1 File.

### mHsp70-ePGK binding under in-cell heat shock is specific and ATP-dependent

For in-cell experiments, ePGK and either mHsp70 or mHsp70K71M are co-expressed in U-2 OS cells (see Methods). Only cells with 1:1±0.1 substrate:chaperone (based on ratiometric fluorescence intensity calibration) were chosen for imaging. ePGK was thermally denatured by a programmed infrared laser-induced temperature ramp (Fig 1A inset). The average of the $E_{FRET}$ curves for mHsp70-ePGK binding for 15–30 cells was calculated for ePGK0-3 (Fig 7A). As expected for ePGK0 with $T_m$ = 48°C, only the onset of binding was observed. In contrast, ePGK2 and ePGK3 showed fully resolved cooperative binding curves with an overall $E_{FRET}$ change of ~8% on average. The $T_0$ for ePGK2-mHsp70 and ePGK3-mHsp70 binding fit to 39.1°C and 37.1°C, respectively. The fitted $T_0$ values are about 3°C lower than the corresponding ePGK *in vitro* thermal denaturation $T_m$ values, as we further discuss below.

In contrast to our *in vitro* binding results, average $E_{FRET}$ binding curves for the ATPase-inactive mHsp70K71M mutant showed only a small signal change, $\leq 2\%$ (Fig 7B). A control in cells transfected with mEGFP and mCherry (Fig J in S1 File) shows a similar 2% change, so the mHsp70K71M result is likely due to a weak interaction of the fluorescent labels. The signal observed with wild-type mHsp70 in cell is at least 4x times higher than with mHsp70K71M. Therefore, the $E_{FRET}$ change observed with mHsp70 is due to productive substrate binding regulated by ATPase activity. In-cell, mHsp70 is indeed a heat shock chaperone for ePGK.

## Discussion and conclusions

Our results show that *in vitro*, mHsp70 neither accelerates ePGK refolding from denaturant nor binds PGK upon heat shock in an ATP-dependent manner. By contrast in-cell, mHsp70 binds ePGK in a cooperative, ATP-dependent manner. These results are summarized in Fig 8, depicting the results discussed above in a 2D grid for easy visualization.

For in-cell binding, the observed binding midpoint temperature $T_0$ is ~3°C lower than the PGK melting temperature $T_m$ *in vitro*. We previously showed that the stability of FRET-labeled PGK1 (fPGK1) is higher in U-2 OS cells by ~2°C than *in vitro* [31,40]. Thus, the Hsp70-PGK binding temperature $T_0$ is actually likely to be ~5°C lower than the in-cell $T_m$ of PGK.

There are two plausible explanations for why mHsp70 already binds ePGK below its in-cell unfolding temperature: mHsp70 either acts as an unfoldase, or as a preemptive holdase for PGK. Unfoldases lower the stability of a protein they bind by shifting the unfolding equilibrium [41], while we coin the term 'preemptive holdase' for a chaperone that recognizes and binds its substrate below $T_m$ without promoting unfolding. Because wt-Hsp70 is overexpressed during the over half hour duration of our previous in-cell fPGK1 thermal denaturation experiments, and a higher fPGK1 $T_m$ is observed in-cell than *in vitro*, these chaperones do not appear to act as unfoldases for PGK in the cell [42].

Instead, Hsp70 may act as a preemptive holdase in the cell. Our proposed mechanism is as follows. It is known that Hsp70 recognizes short (~7 residues) hydrophobic motifs [43]. Increased thermal fluctuations of the native state preceding the thermal unfolding transition have been demonstrated for PGK and other proteins: for example, tryptophan near the protein surface exists in a highly fluctuating environment prior to unfolding [44–46]. We therefore hypothesize that prior to unfolding, native state fluctuations of substrate proteins such as PGK expose on their surface short hydrophobic motifs that Hsp70 recognizes and binds preemptively, before full unfolding occurs. Binding of a human Hsp70 isoform, Hsc70 and bacterial

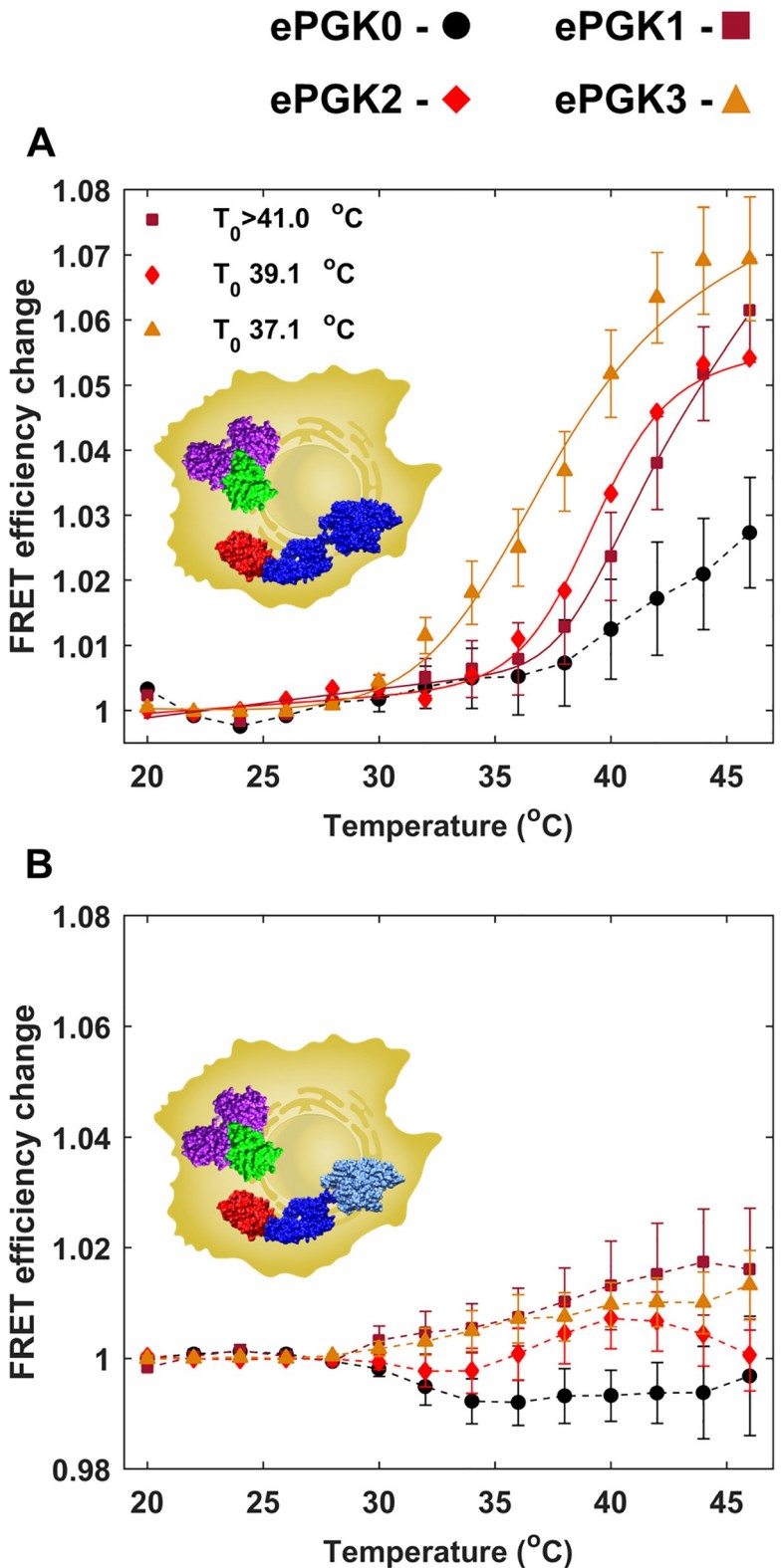

**Fig 7. In-cell binding of mHsp70 and ePGKs.** Filled markers and dashed lines show experimental data and solid lines are fits to experimental data. Error bars are standard error from mean for 15–30 cells. Fit errors reflect the 1 standard deviation precision of the fit. FRET efficiency average for (a) mHsp70- and (b) mHsp70K71M-PGK binding *vs.* temperature for ePGK0 (black), ePGK1 (dark red), ePGK2 (red) and ePGK3 (orange).

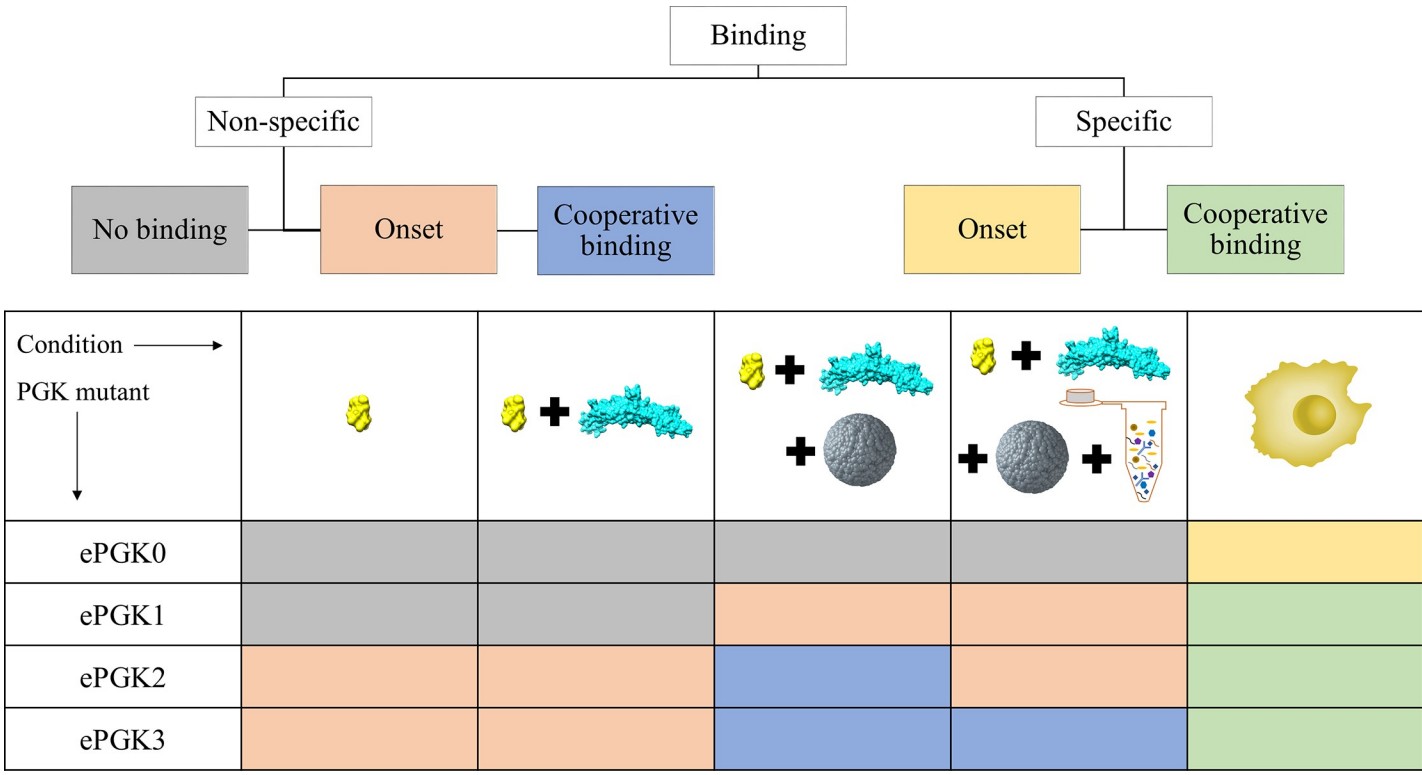

**Fig 8. Summary of the results discussed in the manuscript.** Binding types are shown in the dendrogram on the top. Results are summarized in the grid below. Each unit in the grid is color coded and the color stands for a binding type. Cooperative *and* ATP-dependent specific binding is only observed in-cell. "Onset" means that the beginning of a cooperative binding curve can be observed, but the transition is at too high of a temperature to observe the complete binding curve.

Hsp70, DnaK, to dynamically exposed states was recently demonstrated for two substrate proteins using NMR [47] to support this notion.

PGK is known to have ~7 such hydrophobic motifs that could potentially bind Hsp70, as shown in Fig 9. These binding sites were predicted using the limbo website, http://limbo. switchlab.org/limbo-analysis [48], which was developed for DnaK (bacterial Hsp70), but is still likely useful for other Hsp70s, although it may not properly identify all Hsp70 binding sites in

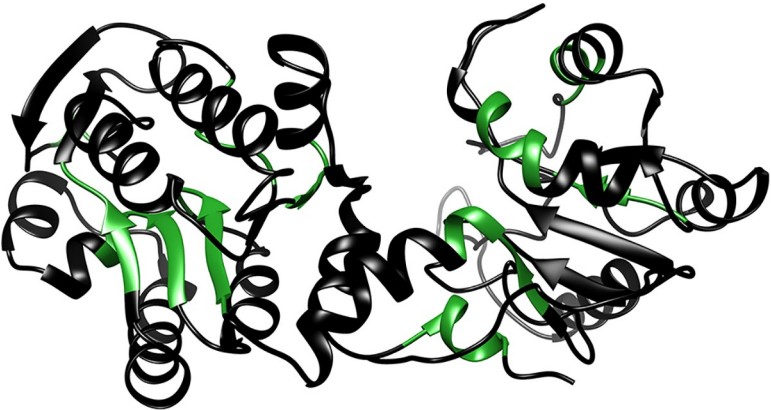

**Fig 9. Ribbon structure of PGK.** Hydrophobic patches are highlighted in green. These patches are potential binding sites for Hsp70 even when PGK is in its native state.

PGK. It has been observed previously that DnaK can bind near-native states of maltose binding protein and prevent unfolding with force [24]. Preemptive holdase activity in the cell similarly would enable Hsp70 to bind and hold substrate proteins prior to unfolding, stabilizing substrates against unfolding. We speculate that preemptive holdase activity may have evolved to preempt formation of misfolded or unfolded states of important substrates before they can have a chance to aggregate in the cell.

In contrast, *in vitro* binding or refolding experiments in buffer or crowder showed equal binding signals of ePGK with both mHsp70 and mHsp70K71M, pointing to an ATP-independent sticking between mHsp70 and unfolded ePGK. We also corroborated that the addition of Hsp40 did not increase binding efficiency *in vitro*. Thus, *in vitro* binding, crowded or not, is not under ATP control and not pre-emptive. However, the addition of cell lysate reduced the propensity for ATP-independent sticking in Ficoll70. We therefore conclude that weak interactions with cytoplasmic components and crowding combine to promote the productive chaperoning interaction of mHsp70 with ePGK under thermal stress. It is worth noting that the FRET efficiency changes observed upon binding in various scenarios (excluding the larger changes at $> 50°C$ due to Hsp70 and/or PGK aggregation) range from +10% to +30%, likely caused by changes in crowding (buffer vs. Ficoll vs. in-cell) and binding sites (specific ATP-dependent vs. non-specific binding) in-cell and *in vitro*.

The observation that yeast PGK is a cooperative and ATP-dependent substrate in-cell but not *in vitro* points to the possibility of Hsp70 having a broader palette of substrates *in vivo* than one would conclude based on *in vitro* assays alone. Chaperoning is likely not the only quinary structure formation process that is significantly modulated by the cytosol. Other weak but functional protein-protein interactions are also likely to elude *in vitro* assays such as pull-downs or titrations, and may require investigation in the native environment of the cell to obtain quantitative $K_d$ values and other binding parameters [14,49–51].

## Supporting information

**S1 File.** Contains supplementary results, supplementary methods, supplementary A through J Figures, and supplementary A through D Tables, highlighting control experiments, protein sequences, and thermodynamic protein data.
(PDF)

## Acknowledgments

The authors thank Prof. Brian Freeman for helpful discussions and suggestions.

## Author Contributions

**Conceptualization:** Martin Gruebele.

**Data curation:** Drishti Guin.

**Formal analysis:** Drishti Guin, Hannah Gelman, Martin Gruebele.

**Funding acquisition:** Martin Gruebele.

**Investigation:** Drishti Guin, Hannah Gelman, Yuhan Wang.

**Methodology:** Drishti Guin, Martin Gruebele.

**Project administration:** Martin Gruebele.

**Software:** Drishti Guin.

**Supervision:** Martin Gruebele.

**Validation:** Drishti Guin.

**Visualization:** Hannah Gelman.

**Writing – original draft:** Drishti Guin, Martin Gruebele.

**Writing – review & editing:** Drishti Guin, Hannah Gelman, Yuhan Wang, Martin Gruebele.

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
