## [Decision Letter · Decision Letter 0]

11 Jul 2019

PONE-D-19-15421

Heat shock-induced chaperoning by Hsp70 is enabled in-cell

PLOS ONE

Dear Dr. Gruebele,

Thank you for submitting your manuscript to PLOS ONE. After careful consideration, we feel that it has merit but does not fully meet PLOS ONE’s publication criteria as it currently stands. Therefore, we invite you to submit a revised version of the manuscript that addresses the points raised during the review process.

Your manuscript has been examined by two experts in the field. The both find your work of interest, but raise a number of issues that need to be addressed in a suitably revised version.

We would appreciate receiving your revised manuscript by Aug 25 2019 11:59PM. To enhance the reproducibility of your results, we recommend that if applicable you deposit your laboratory protocols in protocols.io, where a protocol can be assigned its own identifier (DOI) such that it can be cited independently in the future. For instructions see: http://journals.plos.org/plosone/s/submission-guidelines#loc-laboratory-protocols

We look forward to receiving your revised manuscript.

Kind regards,

Jose M. Sanchez-Ruiz

Academic Editor

PLOS ONE

**Journal Requirements:**

2. Thank you for stating that “NO, The funders had no role in study design, data collection and analysis, decision to publish, or preparation of the manuscript” in your financial disclosure.

Please also provide the name of the funders of this study (as well as grant numbers if available) in your financial disclosure statement.

**Comments to the Author**

1. Is the manuscript technically sound, and do the data support the conclusions?

Reviewer #1: Yes

Reviewer #2: Yes

2. Has the statistical analysis been performed appropriately and rigorously? 

Reviewer #1: Yes

Reviewer #2: Yes

3. Have the authors made all data underlying the findings in their manuscript fully available?

Reviewer #1: Yes

Reviewer #2: Yes

4. Is the manuscript presented in an intelligible fashion and written in standard English?

Reviewer #1: Yes

Reviewer #2: Yes

5. Review Comments to the Author

Reviewer #1: The authors investigate how the molecular chaperone Hsp70 binds a client protein, PGK, during heat shock. Their FRET studies show that while binding is non-specific in the test tube, there is some specific binding in cells that likely occurs as hydrophobic regions in PGK are exposed during heating. This manuscript provides an important glimpse into how the Hsp70 class of molecular chaperones protect proteins from heat shock in cells and should be published after some revisions.

Equations 5 accounts for the unfolded state of the client protein, the chaperone bound client protein and the aggregates of the client-chaperone complex. But, especially under heat shock one might expect the unbound client to aggregate even when not chaperone bound. Is there a reason to exclude this aggregated species from the analysis?

The PGK Construct: The Hsp70 and Hsp40 constructs used for these experiments are the human molecular chaperones, but the PGK construct is from yeast. What is the percent identity between human PGK and the wild-type PGK construct used in these experiments? Also, for figure 8, how were the green "hydrophobic patches" identified? Do these regions overlap with regions that might bind Hsp70 based on the Limbo server (http://limbo.switchlab.org/limbo-analysis)? The Limbo server is based on the binding propensities of the E. coli Hsp70 DnaK, nonetheless, these predictions might be useful in considering what regions of PGK might be involved in binding human HspA1A.

In Figure 2, the authors show that the Hsp70/Hsp40 system can improve PGK refolding efficiency following denaturation in guanidine hydrochloride and they use these data to conclude "Thus, PGK1 refolding from denaturant is chaperoned by Hsp70 and eukaryotic PGK is an Hsp70 substrate." However, they later show that at elevated temperature in vitro interactions between PGK and Hsp70 are non-specific. Thus, it seems possible that the more efficient refolding also arises due to a non-specific interaction. Does the K71M Hsp70 mutant have the same effect? (If this experiment was not performed it is sufficient to replace "is an Hsp70 substrate" with something like "appears to be..")

Page 2 line 59: "ATP hydrolysis is induced by substrate binding….."

This description is not quite correct. Hsp70s have a basal rate of ATP hydrolysis which is increased (stimulated) by substrate binding.

Page 4 lines 106-107 "There is a precedent for such native state binding activity for the Hsp70 homolog DnaK, where the chaperone binds to full-length native proteins in force-unfolding experiments [25]."

This reference is not correct – likely it should be reference 24.

Page 6 Line 179: "Tryptophan was excited at 295 nm, and emission spectra were collected from 290 to 450 nm."

Either the excitation wavelength or emission range is likely incorrect.

Reviewer #2: The authors compare in-vitro and in-cell measurements of heat shock proteins binding to

its substrate and find specific differences. They attribute these differences to in-cell

conditions that promote a 'preemptive holdase' mechanism for the chaperone to bind and not

promote unfolding. The experiments have been performed carefully and the manuscript is

written well. I have the following minor comments:

1) The authors should report the fitting errors for the various parameters including Tm, T0,

fast phase and slow phase time-constants.

2) How are the surface exposed sites identified in Figure 8?

3) It would be good to provide a cartoon that summarizes the result given that multiple proteins

and conditions have been explored.

4) The amplitudes of the unfolding curves are very different. Do they have any mechanistic

origins in terms of binding or folding or aggregation?

6. PLOS authors have the option to publish the peer review history of their article (what does this mean?). If published, this will include your full peer review and any attached files.

Reviewer #1: No

Reviewer #2: No

---

## [Author Response · Author response to Decision Letter 0]

26 Aug 2019

The response to reviewers is also included in the cover letter and a PDF response file.

Reviewer #1: 

The authors investigate how the molecular chaperone Hsp70 binds a client protein, PGK, during heat shock. Their FRET studies show that while binding is non-specific in the test tube, there is some specific binding in cells that likely occurs as hydrophobic regions in PGK are exposed during heating. This manuscript provides an important glimpse into how the Hsp70 class of molecular chaperones protect proteins from heat shock in cells and should be published after some revisions.

Equations 5 accounts for the unfolded state of the client protein, the chaperone bound client protein and the aggregates of the client-chaperone complex. But, especially under heat shock one might expect the unbound client to aggregate even when not chaperone bound. Is there a reason to exclude this aggregated species from the analysis?

Author response: The authors thank the reviewer for the helpful comment. When all our PGK constructs are FRET-labeled (with two labels) so as to probe protein folding, a small signature of aggregation is indeed observed around 50 °C. See figure below, where unfolding leads to a decrease of the FRET efficiency, but a small increase after unfolding is visible at ~52 °C due to aggregation. We have to neglect this state for two reasons: (1) It does not have a significant signal in our experiment because our PGK is only singly-labeled, not FRET-labeled; (2) in the experiments with ATP, the binding occurs below 45 °C, before the PGK aggregation makes a significant contribution. Even though some PGK is likely self-aggregated at high temperatures after unfolding, we could not get well-defined fits by including aggregated PGK states, and hence they have not been accounted for in the simplified equation 5. Line 274-278 of the main text now explains this, as well as Figure S1 (also shown below).

The Hsp70 and Hsp40 constructs used for these experiments are the human molecular chaperones, but the PGK construct is from yeast. What is the percent identity between human PGK and the wild-type PGK construct used in these experiments? Also, for figure 8, how were the green "hydrophobic patches" identified? Do these regions overlap with regions that might bind Hsp70 based on the Limbo server (http://limbo.switchlab.org/limbo-analysis)? The Limbo server is based on the binding propensities of the E. coli Hsp70 DnaK, nonetheless, these predictions might be useful in considering what regions of PGK might be involved in binding human HspA1A.

Author response: The authors thank the reviewer for the helpful suggestion of the limbo website. Yeast PGK and human PGK share 65% sequence identity and 76% sequence similarity (Emboss Needle alignment tool). This line was also added to the main text (lines 345-347). The figure 8 was re-plotted as Figure 9 to show the predictions made using the limbo website and it was added in the results (Line 502-504). 

In Figure 2, the authors show that the Hsp70/Hsp40 system can improve PGK refolding efficiency following denaturation in guanidine hydrochloride and they use these data to conclude "Thus, PGK1 refolding from denaturant is chaperoned by Hsp70 and eukaryotic PGK is an Hsp70 substrate." However, they later show that at elevated temperature in vitro interactions between PGK and Hsp70 are non-specific. Thus, it seems possible that the more efficient refolding also arises due to a non-specific interaction. Does the K71M Hsp70 mutant have the same effect? (If this experiment was not performed it is sufficient to replace "is an Hsp70 substrate" with something like "appears to be..")

Author response: This turned out to be a very important suggestion. We added an additional experiment to figure 3 comparing recovery with wt-Hsp70 vs. the ATPase-deficient mutant Hsp70K71M. PGK is indeed a non-specific substrate for Hsp70 both by heat shock AND by denaturant. Lines 25, 38, 100-109, 367-373, 512 were changed accordingly. We also added a small section under methods detailing the fitting function lines 290-297. We also did the new stopped flow experiments with mEGFP as the donor instead of AcGFP1 as reported previously, to make all in vitro and in-cell data even more directly comparable. Thus Line 147-149 now says mEGFP instead of AcGFP1. Our results show that indeed PGK is a non-specific substrate even by stopped flow. Thus neither in vitro condition, heat shock or denaturant, yielded specific binding, but specific binding was rescued in the cell. We thinks this is an even more exciting result (and hope the reviewer agrees!), as it points to the possibility of Hsp70 having an even wider palette of substrates in-cell than in vitro, as we now point out in the Discussion and Conclusions.

Page 2 line 59: "ATP hydrolysis is induced by substrate binding….." This description is not quite correct. Hsp70s have a basal rate of ATP hydrolysis which is increased (stimulated) by substrate binding.

Author response: The statement was fixed according to reviewer comment.

Page 4 lines 106-107 "There is a precedent for such native state binding activity for the Hsp70 homolog DnaK, where the chaperone binds to full-length native proteins in force-unfolding experiments [25]." This reference is not correct – likely it should be reference 24.

Author response: The reference was fixed.

Page 6 Line 179: "Tryptophan was excited at 295 nm, and emission spectra were collected from 290 to 450 nm." Either the excitation wavelength or emission range is likely incorrect.

Author response: Generally, our samples with ATP are excited at 295 nm to avoid the intense ATP absorbance at 280 nm, and maximize the tryptophan over the tyrosine contribution to the fluorescence. We start collecting emission at 290 nm anyway, which overlaps with the excitation as pointed out by the reviewer. This is done so we can monitor the excitation peak for signs of aggregation: formation of aggregates leads to scatter and a drastic change in excitation signal. If this occurs, the experiments are repeated. This clarification is added to the methods (line 180-184).

Reviewer #2: 

The authors compare in-vitro and in-cell measurements of heat shock proteins binding to its substrate and find specific differences. They attribute these differences to in-cell conditions that promote a 'preemptive holdase' mechanism for the chaperone to bind and not promote unfolding. The experiments have been performed carefully and the manuscript is written well. I have the following minor comments:

1) The authors should report the fitting errors for the various parameters including Tm, T0, fast phase and slow phase time-constants.

Author response: An additional table S4 was added that shows errors for in vitro fits. Both tables S3 and S4 show all fit errors referenced in the main text in line 348 and 445.

2) How are the surface exposed sites identified in Figure 8?

Author response: Originally the authors reported any 7 aa long hydrophobic stretches in the protein. However, following the suggestion of reviewer 1, the website http://limbo.switchlab.org/limbo-analysis was used to predict binding sites. Limbo predicts binding sites for the bacterial Hsp70, DnaK, on a potential substrate. The overall pattern is similar, but the limbo-predicted patterns in the new Figure 9 are probably more accurate than our previous version of Figure 8.

3) It would be good to provide a cartoon that summarizes the result given that multiple proteins and conditions have been explored.

Author response: A new Figure 8 summarizes the overall results discussed in the paper.

4) The amplitudes of the unfolding curves are very different. Do they have any mechanistic origins in terms of binding or folding or aggregation?

Author response: Indeed, the maximum changes in FRET efficiency from Hsp-PGK binding vary between 1.1 and 1.3 – not counting the large changes at > 50° from Hsp-self aggregation. We believe these differences are caused by changes in crowding (buffer vs. Ficoll vs. in-cell) and binding sites (specific ATP-dependent vs. non-specific binding), but currently no structural models are available to quantify this. We discuss it now briefly in the Discussion and Conclusions section (lines 520-522).

---

## [Decision Letter · Decision Letter 1]

12 Sep 2019

[EXSCINDED]

Heat shock-induced chaperoning by Hsp70 is enabled in-cell

PONE-D-19-15421R1

Dear Dr. Gruebele,

We are pleased to inform you that your manuscript has been judged scientifically suitable for publication and will be formally accepted for publication once it complies with all outstanding technical requirements.

With kind regards,

Jose M. Sanchez-Ruiz

Academic Editor

PLOS ONE

Additional Editor Comments (optional):

The authors must ensure that their final version for publication includes the very minor changes suggested by reviewer 1 and that it complies with PLOS policy regarding data availability, as noted by reviewer 1.

Reviewers' comments:

Reviewer's Responses to Questions

**Comments to the Author**

1. If the authors have adequately addressed your comments raised in a previous round of review and you feel that this manuscript is now acceptable for publication, you may indicate that here to bypass the “Comments to the Author” section, enter your conflict of interest statement in the “Confidential to Editor” section, and submit your "Accept" recommendation.

Reviewer #1: (No Response)

Reviewer #2: All comments have been addressed

2. Is the manuscript technically sound, and do the data support the conclusions?

Reviewer #1: Yes

Reviewer #2: Yes

3. Has the statistical analysis been performed appropriately and rigorously? 

Reviewer #1: Yes

Reviewer #2: Yes

4. Have the authors made all data underlying the findings in their manuscript fully available?

Reviewer #1: No

Reviewer #2: Yes

5. Is the manuscript presented in an intelligible fashion and written in standard English?

Reviewer #1: Yes

Reviewer #2: Yes

6. Review Comments to the Author

Reviewer #1: The authors response to the reviewers is excellent and the manuscript should be published after the following minor revisions:

- The Limbo algorithm was developed for the E. coli Hsp70 DnaK. While it is still likely useful for other Hsp70s, it may not properly identify all the Hsp70 binding sites in PGK and the authors should mention this caveat in the manuscript.

- The word "onset" in Figure 8 is a bit non-specific. The figure or figure legend should be modified to better explain the idea behind "onset"

- Some further copy editing might be useful.

Reviewer #2: (No Response)

7. PLOS authors have the option to publish the peer review history of their article (what does this mean?). If published, this will include your full peer review and any attached files.

Reviewer #1: No

Reviewer #2: No

---

## [Editor Report · Acceptance letter]

18 Sep 2019

PONE-D-19-15421R1 

Heat shock-induced chaperoning by Hsp70 is enabled in-cell 

Dear Dr. Gruebele:

I am pleased to inform you that your manuscript has been deemed suitable for publication in PLOS ONE. Congratulations! Your manuscript is now with our production department. 

With kind regards,

on behalf of

Prof. Jose M. Sanchez-Ruiz 

Academic Editor

PLOS ONE